# 3D virtual reconstruction of the Kebara 2 Neandertal thorax

Asier Gómez-Olivencia[1,2,3,4], Alon Barash[5], Daniel García-Martínez[6], Mikel Arlegi[1,7], Patricia Kramer[8], Markus Bastir [6] & Ella Been[9,10]

The size and shape of the Neandertal thorax has been debated since the first discovery of Neandertal ribs more than 150 years ago, with workers proposing different interpretations ranging from a Neandertal thoracic morphology that is indistinguishable from modern humans, to one that was significantly different from them. Here, we provide a virtual 3D reconstruction of the thorax of the adult male Kebara 2 Neandertal. Our analyses reveal that the Kebara 2 thorax is significantly different but not larger from that of modern humans, wider in its lower segment, which parallels his wide bi-iliac breadth, and with a more invaginated vertebral column. Kinematic analyses show that rib cages that are wider in their lower segment produce greater overall size increments (respiratory capacity) during inspiration. We hypothesize that Neandertals may have had a subtle, but somewhat different breathing mechanism compared to modern humans.

[1] Dept. Estratigrafía y Paleontología, Facultad de Ciencia y Tecnología, Universidad del País Vasco/Euskal Herriko Unibertsitatea (UPV/EHU), Barrio Sarriena s/n, 48940 Leioa, Spain. [2] IKERBASQUE. Basque Foundation for Science, 48013 Bilbao, Spain. [3] Équipe de Paléontologie Humaine, UMR 7194, CNRS, Département de Préhistoire, Muséum National d'Histoire naturelle, Musée de l'Homme, 17, Place du Trocadéro, 75016 Paris, France. [4] Centro Mixto UCM-ISCIII de Evolución y Comportamiento Humanos, Avda. Monforte de Lemos, 5, Madrid 28029, Spain. [5] Faculty of Medicine in the Galilee, Bar-Ilan University, Henrietta Szold, 8. P.O.B 1589, 1311502 Zefat, Israel. [6] Paleoanthropology Group, Museo Nacional de Ciencias Naturales (CSIC), J. G. Abascal 2, 28006 Madrid, Spain. [7] Université de Bordeaux, Allée Geoffroy Saint-Hilaire, PACEA UMR 5199, Bâtiment B8, 33615 Pessac, France. [8] Departments of Anthropology and Orthopaedics and Sports Medicine, University of Washington, Seattle, WA 98195-3100, USA. [9] Department of Sports Therapy, Faculty of Health Professions, Ono Academic College, 5545001 Kiryat Ono, Israel. [10] Department of Anatomy and Anthropology, Sackler Faculty of Medicine, Tel Aviv University, 6997801 Tel Aviv, Israel. These authors contributed equally: Asier Gómez-Olivencia, Ella Been. Correspondence and requests for materials should be addressed to A.Góm-O. (email: asier.gomezo@ehu.eus)

The study of the evolution of the human thorax is paramount to understanding key aspects of human paleobiology. The thorax is the insertion point of muscles related to the movement of the upper limb, connects the latter to the trunk[1], and protects important organs of the chest. The general size of the thorax of organisms has been correlated with their total lung capacity, an important physiological variable[2–4]. The costal skeleton and the spine, which provide important clues for understanding locomotion and posture[5–7], are interdependent. The study of the thorax is hampered, however, by the intrinsic difficulties of studying metameric elements, the fragility of both ribs and vertebrae and their scarcity in the fossil record[8–10].

The size and shape of the Neandertal thorax has been a subject of scientific debate for more than 150 years. Fuhlrott[11] described the costal remains belonging to the Feldhofer 1 Neandertal individual found in 1856 as having a circular cross-section and a low degree of curvature. Similar anatomical features were proposed for the fossils found in Spy (Belgium)[12], Krapina (Croatia)[13], and La Chapelle-aux-Saints 1 (France)[14]. Gorjanović-Kramberger[13] suggested a dorso-ventrally expanded thorax based on the low degree of curvature of the first ribs of Krapina, which was also proposed based on the conoid length of the Neandertal clavicles[15]. The thorax of the Tabun C1 female Neandertal skeleton was estimated to be large (relative to stature) with very curved and more horizontally oriented ribs than modern humans[16]. However, other studies proposed that Neandertal ribs were very similar to that of modern populations[17] or just observed that Neandertal ribs were more robust, but were not, on their own, evidence for a more voluminous thorax compared to modern humans[18]. The reassessments of the costal skeletons of Shanidar 3[8], Kebara 2 (K2)[9,10], La Chapelle-aux-Saints 1[19], and the description of new Neandertal costal remains, from the site of El Sidrón[20,21] and Regourdou 1[22] individual provide new information. Additionally, the implementation of 3D geometric morphometric techniques to these studies helps to quantify differences between the modern human and Neandertal thorax[10,20,21,23].

The larger costal skeleton in Neandertals, with longer mid-thoracic ribs than modern humans, was hypothesized to result in a more voluminous thorax. This larger thoracic volume could have been related to a need for more oxygen intake due to their larger body masses and hunter-gatherer life-style[4,8,24], and an exaptation for cold-climate conditions[9]. The larger Neandertal costal skeleton, especially the longer mid-thoracic ribs, has also been linked to longer mesosterna[25]. The presence of less-curved first ribs has been confirmed in Neandertals and has been linked to shape differences in the upper thorax[20–22]. Finally, the more dorsally oriented transverse processes in the mid-thoracic spine has been related to a larger degree of invagination of the spine within the thorax[26]. All the proposals relative to the general size and shape of the Neandertal thorax were derived from an analytic approach based mainly on isolated skeletal specimens, pointing towards important differences between the Neandertal thorax and that of modern humans.

To our knowledge, only one study has approached the problem of the size and shape of the Neandertal thorax from a synthetic approach. Within the context of the reconstruction of a complete Neandertal skeleton, skeletal elements from La Ferrassie 1 were combined with other Neandertal individuals, including the thoracic and lumbar spine and costal skeleton of K2[27]. The first two ribs of K2 were elongated in order to fit the thorax to the longer clavicles and generally larger La Ferrassie 1 body frame[27]. Sawyer and Maley[27] also indicated the presence of flaring lower thorax which resulted in what they called a "bell-shaped thoracic region". More recent studies have provided important morphological information regarding the spinal morphology of

Neandertals which has resulted in an interpretative shift regarding Neandertal posture[6,7,28–31]. Additionally, errors in the reconstruction of the K2 ribs have been documented[9,10], which could have affected the previous reconstruction of this thorax.

Here we present the 3D virtual reconstruction of the most complete adult Neandertal thorax found to date, that of the K2 individual. The cave site of Kebara is located on Mount Carmel, 13 km south of Wadi el Mughara[32]. This site has yielded two partial Neandertal skeletons: the 8–9 month infant Kebara 1[33] and the adult K2[34], as well as other isolated Neandertal remains[35]. The burial of K2 was found in 1983 in level XII[36], which is dated by thermoluminiscence in $60 \pm 3.5$ ky[37]. The skeleton preserves the mandible, an upper third molar, the hyoid, the scapular girdle and the upper limb, the pelvic girdle, the proximal half of the left femur, and the most complete Neandertal vertebral column and thorax found to date[34,36] (Supplementary Figs. 1–5). The average age-at-death of K2 based on different methods is 32 years[38]. K2 has been determined to be male based on pelvic morphology[36,39] and his stature was estimated between 168.7[40] and 170.3 cm[6]. His body mass was estimated at 75.6 kg[41]. The skeleton of K2 shows several anomalies and/or pathological lesions including the presence intercostal ossifications and pseudoarthroses in ribs 5–7 from the right side[42]. Recent research has, however, ruled out previous hypotheses, pointing out that the apophyses and nearthrosis observed in the ribs are caused by a genetically driven anatomical variant[43], and there is no evidence to assume that this anatomical variant implies any functional constraint in the ribcage. We have also detected the presence of a slight scoliosis, which did not reach a pathological level[6]. The 3D virtual reconstruction of the K2 thorax provides a comprehensive context for the differences found in the isolated vertebrae and ribs of Neandertals when compared to modern humans. With this reconstruction, we are able to perform a comparative morphometric analysis of the K2 thorax which indicates that the differences in vertebrae and ribs result in differences in the thorax as a whole between modern humans and Neandertals. We also discuss these differences of the thorax in relationship to the Neandertal lumbo-pelvic complex and its evolutionary implications. This virtual reconstruction of an extinct hominin trunk reveals the interdependence of the pelvic morphology, the spine, and the costal skeleton.

## Results

**Size and shape of the 3D reconstruction of the Kebara 2 thorax.** The 3D virtual reconstruction of the Kebara 2 Neandertal trunk is shown in Fig. 1. Measurements of the thorax of Kebara 2 compared to our modern male sample are shown in Table 1. Kebara shows a mild (i.e., <20°) scoliosis[6] and a genetically driven anatomical variant in ribs 5–7 from the right side[42,43]. We compared our 3D reconstruction of the K2 thorax to the mean shape of our modern male sample. In order to facilitate the visualization of this comparison, a modern thorax was warped into the K2 shape and then superimposed to the modern male mean (Fig. 2). In cranial view, the most striking feature is the invagination of the thoracic spine into the K2 thorax: in K2 the dorsal-most tips of the spinous processes are embedded within the limits defined by the dorsal surfaces of the posterior angles of the ribs, while in modern humans these tips project more dorsally. This invagination is consistent with previous suggestions based on the orientation of the transverse processes of the thoracic vertebrae[26] (Supplementary Note 1).

In ventral view, when compared to modern humans, K2 shows a wider mid-lower thorax (mainly around ribs 6–9). This is in accordance with the maximum width of the thorax, which is larger in our reconstruction than in our modern male

comparative sample (Table 1). In lateral view, the ribs of K2 show a more horizontal orientation, which results in a slightly more antero-posteriorly deep thorax (Table 1). In this view, it is also possible to see the less marked kyphosis of the thoracic spine[6]. We should note that, despite not being significantly different, K2 shows a cranio-caudally shorter thorax than our modern human comparative sample (Table 1).

The geometric morphometric analyses of the whole thorax in shape space[44] (Fig. 3; Supplementary Figs. 8, 9), i.e., without the influence of size, demonstrates that the thorax morphology of the K2 individual falls outside modern human variability when we plot PC2 against PC3 (Fig. 3). These differences were confirmed using a permutation test: K2 is significantly different from the modern human male mean (Procrustes distance = 0.1098; $p <$

0.01). These morphological differences are mainly related to the wider mid-lower thorax of K2 compared to that of modern human sample.

Finally, despite the differences found in the width and depth of the K2 thorax (Table 1), the overall size (based on the centroid size) of the K2 reconstructed thorax is similar to that of the modern human comparative sample (Table 1). This results from the shorter cranio-caudal dimension of the thorax and from the deeper invagination of the vertebral bodies in K2, when compared to modern humans. In summary, our reconstruction of the K2 thorax shows that, when compared to modern human males of similar stature (using thorax height and humeral maximum length as a predictor of stature), this individual showed a significantly wider mid-lower thorax which was slightly larger dorso-ventrally. Additionally, our reconstruction shows that the thoracic spine was invaginated within the K2 thorax to a larger degree than in modern humans and that his ribs were more horizontally oriented. Despite these significant morphological differences, the thorax of Kebara 2 was not significantly larger than that of modern humans.

## Discussion

Here we provide the accurate reconstruction of the most complete adult Neandertal thorax found to date: that of Kebara 2. Our reconstruction shows significant metric and morphological differences when compared to modern human males of similar stature. These differences can be explained by differences in the morphology of the thoracic spine, the costal skeleton, and the interplay between these two anatomical regions. Further, we hypothesize that the pattern shown by the K2 reconstruction can be extended to other Neandertals for two reasons: first, the restricted hypodigm of Neandertal mid- and lower thoracic vertebrae shows more dorsally oriented transverse processes, evidence for a more invaginated spine[26] (Supplementary Note 1); and second, all the Neandertal individuals on which it has been possible to perform metric comparative analyses of their costal skeleton, show the same metric and/or morphological differences with modern humans[8–10,19–22].

The size of the Neandertal thorax has been a matter of debate due not only to its relationship to the general skeletal structure, but also due to its relationship to the total lung capacity[4,24]. Previous studies have emphasized the larger size of the K2 costal skeleton based on the longer ribs of the mid-thorax[9,10]. In this study, we provide evidence of a larger degree of invagination of the vertebral column into the thorax in Kebara 2. We consider that some of the differences observable in the Neandertal costal skeleton, such as a longer distance between the tubercle and the posterior angle, is related to this invagination. The posterior angle marks the insertion point of the erector spinae muscles, and a

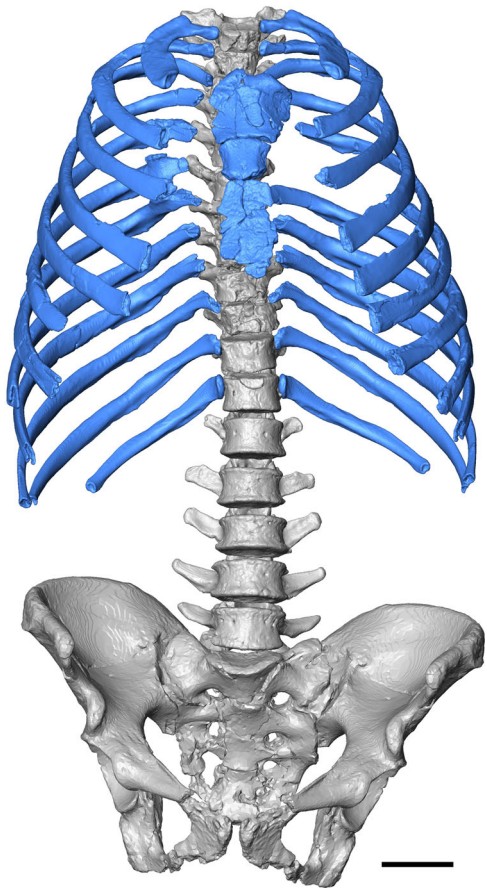

**Fig. 1** Ventral view of the reconstructed thorax of Kebara 2. The blue color is to highlight the ribs and the sternum. Scale bar = 5 cm

| | Sex | Age/age-at-death | Full thorax CS | Maximum thorax width (mm) | Thorax depth[a] (mm) | Thorax height[b] (mm) | Humeral length (mm) | Stature estimation[c] (cm) | Bi-iliac breadth (mm) |
|---|---|---|---|---|---|---|---|---|---|
| Kebara 2 | Male | 25–39[38] | 3281.19 | 326.8[d] | 224 | 265.5 | 317(R)/324 (L)[e] | 165.45 | 319.7[d]/313[52,d] |
| *Homo sapiens* | Males (n = 16) | 70.63 ± 10.04 | 3303.95 ± 112.47 | 298.16 ± 13.87 | 217.54 ± 13.95 | 282.85 ± 19.38 | 320.34 ± 12.38 (n = 14) | 167.00 ± 5.72 (n = 14) | 284.13 ± 12.15 |

**Table 1 Full thorax centroid size (CS), width, depth, and other general measurements of Kebara 2 compared to the male modern human sample**

[a]Antero-posterior dimension of the thorax at the fifth thoracic vertebral (T5) level (between the tip of the spinous process of T5 and the line connecting the sternal ends of the 5th ribs)
[b]Measured in the mid-sagittal plane, distance from the cranial-ventralmost point of the vertebral body of T1 to the caudal-ventralmost of the vertebral body of T12
[c]Following Sjøvold[74] and Carretero et al.[40] using humeral length
[d]Significantly different from the modern male comparative sample
[e]Humeral length data from Vandermeersch[75]

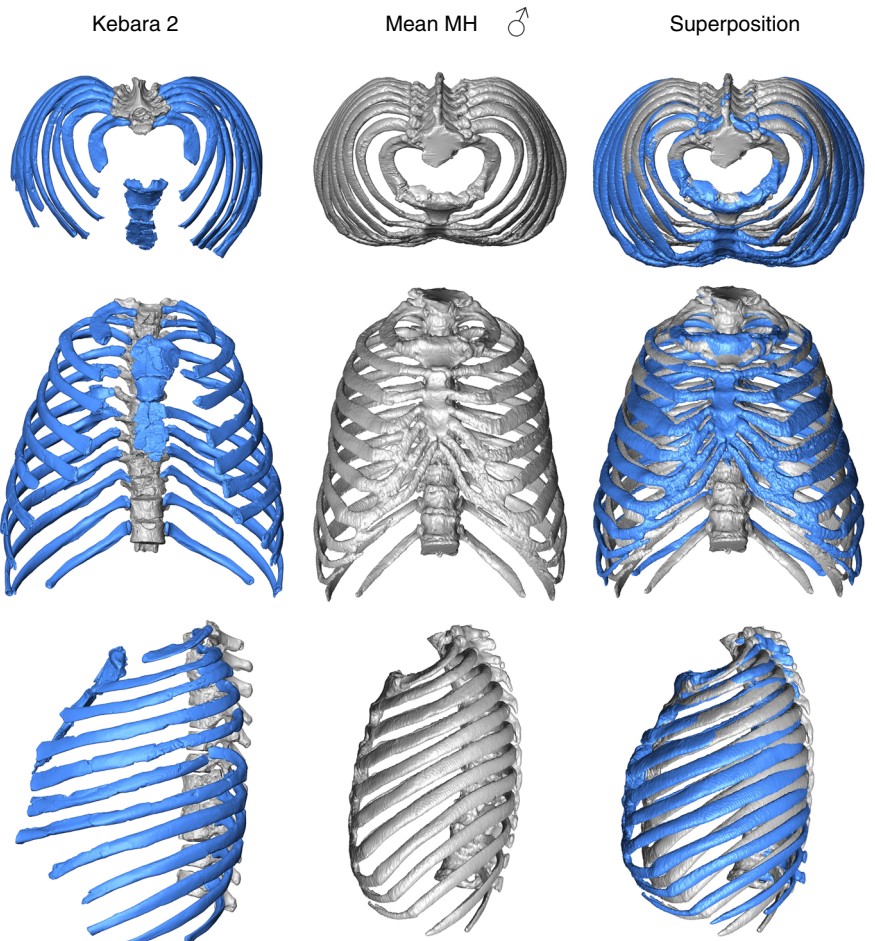

**Fig. 2** Comparison of the Kebara 2 (K2) thorax (left column; see legend in Fig. 1 for color explanation) to the modern human male sample (in gray; middle column) and superposition of the two morphologies (in blue, Kebara 2; in gray, the modern human sample male mean; right column) in cranial (top row), ventral (middle row) and left lateral (lower row) views. In the cranial view, the invagination of the K2 spine into its thorax is noticeable. In ventral view, the relatively (and absolutely; see Table 1) wider thorax of Kebara 2 compared to modern humans is appreciable. In lateral view, K2 shows relatively larger antero-posterior mid-thorax and straighter shafts of the rib, while modern humans show more caudally curved ribs sternal to the posterior angle. In the third column the same template has been used in order to represent the modern human male mean (gray) and the K2 morphology. In the superposition, both the modern human male thorax and the K2 thorax scaled to the same centroid size

more invaginated thoracic spine would require not only more dorsally oriented transverse processes, but also longer shafts in the segment between the posterior angle and the tubercle of the ribs[9]. While this would result in an absolutely longer rib, which is the case of the K2 mid-thoracic ribs[9,10], it does not affect to the overall size of the thorax. This is due to the fact that the general size of the thorax is an interplay of not only the size of the costal skeleton, but also its articulation with the spine[21].

Despite the calculated larger energy expenditure of Neandertals[4,24], which was previously proposed as consistent with their larger total lung capacities, our reconstruction does not show a larger skeletal thorax. In some high altitude human populations, an association between ventilatory capacities and thoracic dimensions exists[4,15]. Bellemare et al.[3] indicated, however, that the size of soft tissues provides a better correlate to the lung ventilatory capacities than skeletal measurements, such as the antero-posterior and mediolateral diameters of the thorax. Based on the skeletal morphology, we cannot rule out that the total lung capacity of Neandertals was not different from that of modern humans. Differences in the soft tissues (such as height of the diaphragm)[3] could, however, have resulted in a larger total lung capacity in Neandertals than in modern humans, despite a thorax of overall similar size.

Based on the relationship between the conoid length of the clavicle and the chord of the second rib, Neandertals have an antero-posteriorly expanded thorax[15]. Our K2 thorax reconstruction is slightly more expanded antero-posteriorly than the male comparative sample, although this difference is not significant. Other Neandertal individuals, normally considered as males, or of similar general size to K2, show longer and straighter first ribs (e.g., Regourdou 1 or the partially preserved Amud 1)[9,22]. Thus, it would not be surprising if other Neandertals had larger thoraces in the antero-posterior dimension. In fact, the less-curved first ribs in Neandertals, independently of their absolute size, have been related, through a 2-block partial least squares (PLS) analysis of the first ribs and the rest of the ribcage, to relatively wider thorax in its mid-lower part and to more horizontally oriented ribs in lateral view[20]. This accords with our K2 thorax reconstruction. In the case of Neandertals, this wider, in absolute terms, mid-lower thorax (already noted in a previous reconstruction)[27], would be consistent with the presence of wider pelves in Neandertals compared to modern humans[45,46] (Table 1).

The increase in lung volume during inhalation is mainly related to two mechanisms: diaphragm flattening and the "bucket-handle" and "pump-handle" movements of the ribs[1]. In

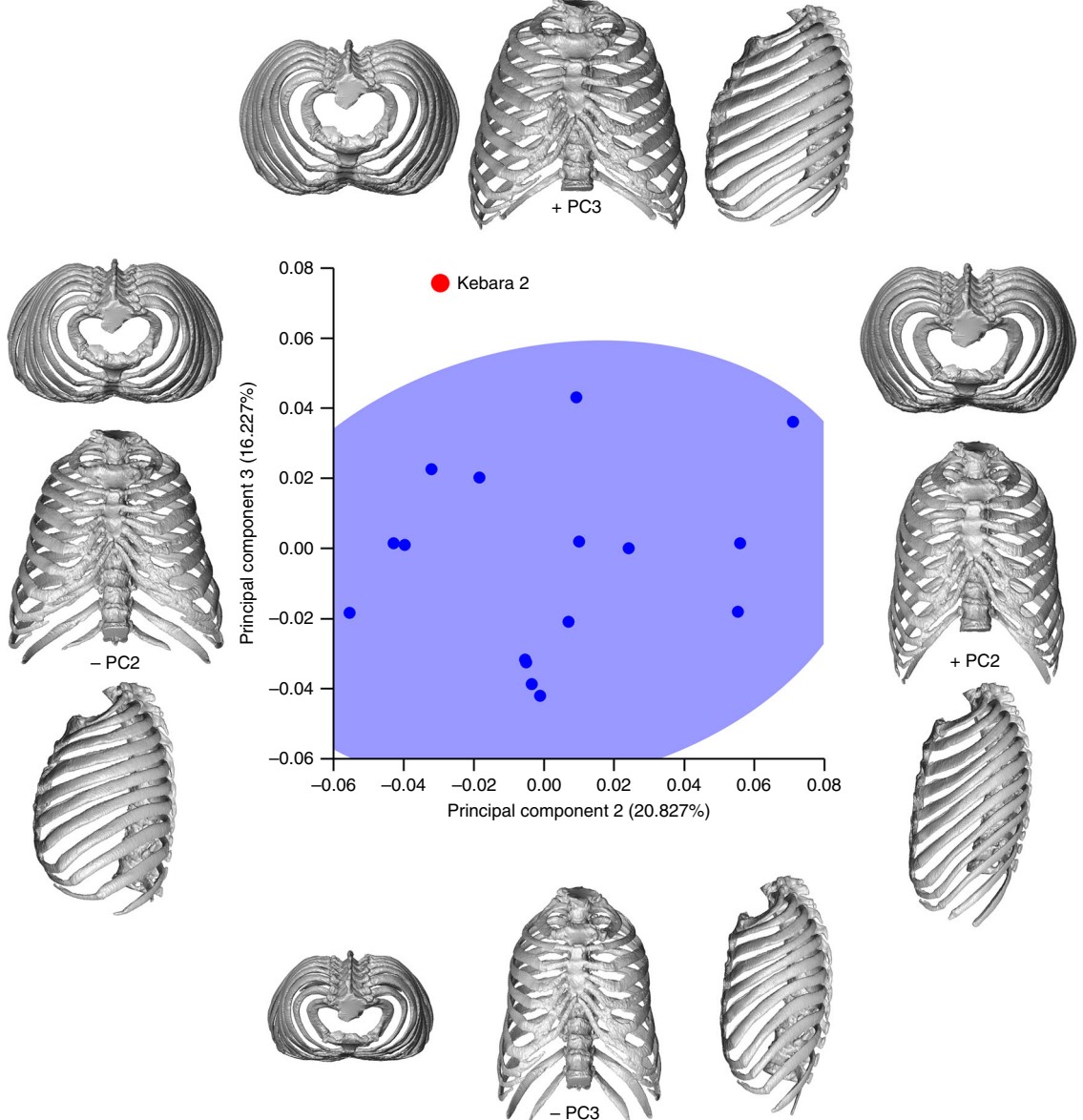

**Fig. 3** Principal component analysis (PCA) representing the second (PC2) and third (PC3) principal components, which represent the 20.827% and the 16.227% of the variation, respectively. In this plot Kebara 2 is outside the 95% of the equiprobability ellipse representing the modern human sample

ribs 1–7, it seems that the "bucket-handle" and "pump-handle" movement of the ribs occur similarly in each level[47], but the "bucket-handle" movement seems to predominate in the lower thorax. Despite similarities in the overall thoracic (centroid) size, K2 likely had a larger surface of the diaphragm due to the significantly larger mediolateral and slightly larger antero-posterior diameters of the lower thorax. On the one hand, the position of the diaphragm was likely a major factor to determine the total lung capacity of Neandertals (as discussed above), but additionally, its larger surface, due to the larger cross-section of the lower thorax, could have enhanced the ability to increase and decrease the total lung volume during breathing. In modern humans, the enlargement of the lower part of the ribcage, i.e., the area that supports the diaphragm, increases considerably the respiratory capacity[48]. In fact a lung volume of about 9.04 l has been calculated for the Kebara 2 individual based on the relationship between total lung capacity and costal arc length[49].

On the other hand, in lateral view, the K2 reconstruction shows more horizontal ribs than is seen in modern humans. This likely constrained the rib elevation in the sagittal plane, which is related to the bucket-handle movements of the ribs. The larger articular tubercles of the lower ribs of Neandertals when compared to modern humans[8,21,22], could be related to strong breathing kinematics in the lower thorax, which could also be related to the function of the diaphragm. In fact, kinematic analyses show that rib cages wider in its lower segment produce greater overall size increments (respiratory capacity) during inspiration[50]. In summary, we hypothesize that Neandertals may have had a somewhat different breathing mechanism, one which relied relatively more on diaphragm contraction, than is exhibited in modern humans. Consideration of the potential differences between Neandertals and modern humans in hematologic and biochemistry that could potentially also significantly affect the respiratory physiology, as has been seen in different extant modern human groups[51], is beyond the scope of this report.

To fully comprehend the Neandertal thorax anatomy, understanding its relationships with the adjacent anatomical regions is critical. The reconstructed thorax presented here allows us to

understand the biomechanical implications of the observed differences in the Neandertal thoracic and lumbar spine and the individual differences that we have documented in the costal skeleton. We hypothesize that, in Neandertals, the orientation and position of the sacrum within the pelvis is not only responsible for (or, at least, related to) the lower degree of curvatures of the spine, but also explains the invaginated spine within the thorax. The Neandertal sacrum is more vertically oriented than in modern humans, which is connected to the lower degree of lumbar lordosis[6,7,28,29,31]. At the same time, the Neandertal sacrum is positioned more ventrally relative to the dorsal end of the iliac tuberosities of the pelvis, than that of modern humans[52,53]. The dorsal projection of the iliac tuberosities would mark the dorsal end of the trunk (encompassing the pelvis, spine, and costal skeleton). Thus, a relatively more ventral sacrum within the pelvis could result in the invagination of the spine within the thorax, which would also affect the orientation of the transverse processes of the thoracic vertebrae and the length of the rib shafts between the tubercle and the posterior angle in the mid-thoracic ribs[9] (as discussed above). A more invaginated spine would reduce the inertia moments of the costal skeleton with regards to the spine. Moreover, the orientation of the lumbar transverse processes, which are more laterally and vertically oriented than in modern humans[31,54], would provide advantage in mediolateral flexion (e.g., the action of *M. quadratus lumborum*), which would be useful to stabilize the larger inertia moments of the significantly wider lower thorax of Kebara 2. In parallel, the wider pelvis of Neandertals, when compared to modern humans, likely imposed developmental constraints to the lower part of the thorax[55]. We hypothesize that the narrower lower thorax present in modern humans is likely a derived condition within genus *Homo*, which appeared with the emergence of narrower pelves in *Homo sapiens*[56–59].

Neandertals are significantly different from modern humans in all the spinal regions, and based on comparisons with earlier hominins, previous work has demonstrated that the distinctive features of Neandertals in the vertebral column and pelvis: e.g., lower pelvic incidence, more vertical sacra, lower degree of lumbar lordosis, and lateral orientation of the transverse processes in the mid-lumbar vertebrae, are derived within genus *Homo*[28–31,54,60]. Moreover, some of these features were already present in the Middle Pleistocene population from Sima de los Huesos (SH)[53,61]. A relatively more ventral positioned sacrum was likely present in this Middle Pleistocene population (see Fig. 2 from Bonmatí et al.[53]) which would suggest that, in this population, the invagination of the spine seen in Kebara 2 was already present in the Middle Pleistocene populations ancestral to Neandertals. In the case of the costal skeleton, the only complete first rib from SH is larger than the largest complete Neandertal first rib (Regourdou 1)[22] and the SH hominins show wider pelves than Neandertals. Thus, it would be reasonable to expect larger thoraces in this population than that reconstructed here. The most complete first rib from SH seems, however, to be more curved than Neandertals[62], which would be in accordance to the presence of some, but not all the Neandertal derived traits in this population[61,63]. Unfortunately, the lack of relatively complete Early Pleistocene adult costal remains and the immature status of the only *Homo erectus* costal skeleton (KNM-WT 15000)[64,65] do not provide an evolutionary framework for the evolution of that thorax as complete as that present for the evolution of the vertebral column. In any case, we consider it likely that the modern human thorax morphology is also derived when compared to their Middle Pleistocene ancestral populations.

In summary, the present reconstruction demonstrates that subtle, but significant differences exist in the thorax shape within genus *Homo*. The thorax morphology seems to be the result of the interdependence of several features: general body size and pelvic and spinal morphology. While differences between upper and lower thorax exist[23], some of the elements related to this interplay (and even within the same thorax) may change in a mosaic fashion[66]. Additional fossils and more integration studies are necessary to provide additional evidence to understand the evolution of this anatomical region.

## Methods

**Fossil material**. We have studied the original skeleton of Kebara 2 (K2; housed at Tel Aviv University) and have also used CT-scans of this individual. The scans of the vertebrae and ribs of this individual were made with a medical CT scan at Mount Carmel Medical Center, Haifa (voxel size: $0.598958 \times 598,958 \times 0.5$ mm), and the sacrum and coxal bones were scanned at Sheba Medical Center, Ramat Gan, using a medical CT scanner (slice thickness of 0.625 mm). From these CT-scans, 3D virtual objects of each of the skeletal elements of the trunk (vertebrae, ribs, manubrium and mesosternum) as well of the pelvis were created using Avizo (v. AvizoLite).

**3D virtual reconstruction**. The virtual reconstruction of the thorax was based upon a slightly modified 3D reconstruction of the spine of this individual (Supplementary Note 2) to which the ribs and the sternal elements were added using Avizo (v. AvizoLite). Coxal bones were added to provide a context for the thorax reconstruction.

The ribs of Kebara 2 individual suffer from both taphonomic deformation and error in initial reconstruction. A careful assessment of all the original vertebrae and ribs was performed in order to select the best-preserved ribs and to compensate for potential taphonomic/reconstruction problems. In some cases, mirror-imaging of the skeletal elements was necessary due to taphonomic, reconstruction, and/or preservation concerns (Supplementary Note 2).

**Traditional and geometric morphometric analysis**. The morphometric analysis of the 3D reconstructed thorax was performed using both traditional and geometric morphometric (GMM) analyses. For both the traditional and the GMM analyses, the K2 reconstructed thorax was compared to a sample of medical scans comprising 16 adult modern male thoraxes: 10 of them were scanned for post-mortem autopsy and six were patients at the Ziv Medical Center, Faculty of Medicine, Bar Ilan University (Safed). All individuals were scanned in supine position and we avoided CT-scanning individuals with significant morphological deformations and/or causes of death that might have altered ribcage morphology. Prior to analysis, all CT-data were anonymized to comply with the Helsinki declaration[66,67]. Despite the differences in age (or age-at-death) between K2 and this modern human sample (Table 1), we consider this comparative sample to be representative of modern adult males, because a comparison of this sample to that of another adult modern male sample[50] ($n = 18$; age $= 51.9 \pm 1.2$ years) using a landmark protocol of 402 landmarks (ribs 1–10) did not yield statistical differences between the samples (permutation test; $n = 100$; $p = 0.14$). The traditional morphometric analysis consisted of: (a) the direct comparison of the morphology of the thorax between our reconstruction and modern humans and (b) the comparison of selected measurements between the K2 reconstruction and the modern human comparison sample (Table 1). For the GMM analysis we digitized 526 3D landmarks and semilandmarks on each ribcage using Viewbox 4 software (www.dhal.com), based on a recently published protocol for quantifying the ribcage of hominoids[68]. Additionally, a metric analysis of selected linear measurements of T1–T10 vertebrae was performed using a different comparative sample (Supplementary Note 1).

To assess the size and shape of the thorax using GMM, for ribs 1–10, 20 landmarks (7 fixed and 13 sliding); in ribs 11–12, 19 landmarks (6 fixed and 13 sliding) were captured in each rib, and four fixed landmarks at each thoracic vertebra[68]. For ribs 1–10, fixed landmarks at the rib head were digitized at the most superior and most inferior points with another landmark at the most medial point of the rib head at the inter-articular crest. One additional landmark was placed at the most lateral point of the articular tubercle, and on the shaft, one landmark was placed at the most inferior point at the posterior angle and one each at the most superior and most inferior points of sternal ends. On the 11th and 12th ribs, only six fixed landmarks were collected due to the absence of the inter-articular crest at the rib head. Thirteen equidistant semilandmarks were placed along the lower costal border between the articular tubercle and the inferior sternal end. Additionally, four landmarks were placed at each of the thoracic vertebrae (T1–T12): most antero-superior and most antero-inferior points on the vertebral body at the midline and the most superior and most inferior points at the lower part of the spinous processes. Semilandmarks were slid twice, the first time to the template during digitizing and a second time to the sample average following standard sliding protocol[23,69,70].

In the case of the K2 thorax, we needed to estimate the missing landmarks, mainly at several proximal and distal ends due to the preservation of the fossils (Fig. 1, Supplementary Note 2). Those landmarks were estimated following reference-based estimation through a thin-plate spline (TPS) method[71]. According

to this method, the coordinates of the reference individual, where all landmarks and semilandmarks are present, is used to calculate the missing data of the target specimen through an interpolation function using the reference information, so that the bending energy is minimized[71]. In order to be conservative and due to the lack of other Neandertal rib cages to use as references, we used the coordinates of the mean *H. sapiens* male thorax as a reference specimen. Average *H. sapiens* coordinates were calculated using MorphoJ software[72]. The semilandmarks were slid twice and the full set of 3D coordinates was subjected to a generalized Procrustes analysis[44]. Size was quantified as centroid size (CS) defined as the square root of the summed squared distances of each landmark to the centroid, with the centroid of a configuration calculated as the average coordinate (x, y, z) of the whole set of coordinates[44,73]. Morphological variability of the sample was explored by principal component analyses in shape space and form space[74] using EVAN toolkit software. Mean comparisons were also carried out between K2 and modern human males using a permutation test ($n = 1000$ permutations) in MorphoJ software[72].

## Data availability

Kebara 2 original fossils are accessible with permission of Israel Hershkowitz. The 3D models of the spine and thorax reconstruction of Kebara 2 are available from the corresponding author upon request and in figshare with the identifier https://doi.org/10.6084/m9.figshare.7012256

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

## Acknowledgements

We thank the excavation and research team of Kebara for their work that has allowed for the recovery of the Kebara 2 individual and B. Arensburg, Y. Rak, B. Vandermeersch, O. Bar-Yosef and I. Hershkowitz their permission to study the K2 specimen and V. Slon for technical help. Thanks to our colleagues from MNHN, TAU, EHU-UPV, MNCN, UCM-ISCIII, UCAM, and especially A. Balzeau, J. Stock, C. Shaw, T. Holliday, R.G. Franciscus, T. Durden, J.L. Arsuaga, and J.M. Carretero for their support and constructive discussion. We thank J. Trueba and Madrid Scientific Films for letting us to use one of their photographs. A.G.-O. had a Marie Curie-IEF (FP7-PEOPLE-2012-IEF 327243) research fellowship during part of this work. A.G.-O. and M.A. received support from the Spanish *Ministerio de Ciencia y Tecnología* (Project: CGL-2015-65387-C3-2-P, MINECO/FEDER) and are also part of the Research Group IT1044-16 from the Eusko Jaurlaritza-Gobierno Vasco and Group PPG17/05 from the Universidad del País Vasco-Euskal Herriko Unibertsitatea (UPV/EHU). D.G.-M. and M.B. received support from the *Ministerio de Ciencia y Tecnología* (Project: CGL2015-63648-P).

## Author contributions

A.G.-O. and E.B. designed research; A.G.-O., A.B., D.G.-M., M.A., P.K., M.B. and E.B. performed research; A.G.-O., A.B., D.G.-M., M.A., P.K., M.B. and E.B. analyzed data; A.G.-O., A.B., D.G.-M., M.A., P.K., M.B. and E.B. wrote the paper.

## Additional information

**Competing interests:** The authors declare no competing interests.

