## [Peer Review File · Nature Communications]

Reviewers' Comments:

Reviewer #1:

Remarks to the Author:

In this study, Gómez-Olivencia and colleagues provide a thoughtful and meticulous 3D virtual reconstruction of the thorax of the Kebara 2 Neandertal, the specimen that retains the most complete and best preserved set of thoracic elements yet found for Neandertals, and arguably for all of premodern fossil Homo. A spate of new information and new studies on Neandertal thoracic morphology over the past seventeen years has combined traditional measurements and analysis with new tools like 3D geometric morphometrics to significantly enhance our understanding of this important anatomical region. In fact, during this time we have come to understand the paleobiological implications of Neandertal thoracic anatomy to a far greater degree than the preceding 142 years since the earliest mention of Neandertal rib morphology and their apparent differences from those of extant humans.

This study by Gómez-Olivencia et al. continues this recent spate of new studies and provides both a confirmation of several key functional and biobehavioral ideas previously postulated, as well as some novel aspects. The field of paleoanthropology frequently gravitates towards overemphasizing novel and overstated discoveries that often do not hold up to subsequent analyses. It is refreshing to encounter in this area of research (hominin thoracic function and evolution) a steady accumulation of knowledge that has incrementally refined our knowledge through careful analyses that emphasize important details, and attention to subtle, and critical nuanced information. Gómez-Olivencia and colleagues have been a big part of this successful endeavor, and in this contribution, they continue that trajectory.

The manuscript is well organized and very well written. The importance of the work is nicely contextualized, and the methodological and analytical portions of the study are laid out in sufficient detail to be followed and evaluated. The results are derived logically from the carefully presented data, and are also nicely integrated into the larger set of issues and questions that are driving the study. Beyond the occasional editing that I've noted directly on the manuscript, there is very little to quibble with here.

The one aspect of the manuscript that I think can be improved has to do with the presentation and details provided of the extant human sample from which, and against which the Kebara thoracic remains are compared. Obviously, the analysis depends heavily on the degree to which this sample of males and females accurately reflects average modern human thoracic anatomy. Therefore, several questions arise that are not addressed in the materials as presented:

1. What are the ages of these individuals? Are they all elderly? Or, do they represent a larger range of adult ages? This is potentially important since advanced age can significantly alter prime-age thoracic anatomy due to several hard and soft-tissue changes.
2. What do we know about the overall health of these individuals? Could any of them have had health issues that could have potentially affected their thoracic anatomy?
3. Even in the absence of obvious pathology or extreme age-related changes (if that is the case), on what basis do we know that these 10 males and 10 females reflect average human thoracic size and shape dimensions? Some large research universities with medical schools in which pulmonary health and physiology is studied have developed, or have access to CT-scan studies that have provided a sample of "normals" against which their

clinical patients can be compared. The authors of the current manuscript have done a lot of work, and I am not asking them to re-do their analysis. I am just wondering if there is a way for them to compare their particular sample to a set of such normals, even in a cursory fashion with a few key variables from the published literature, to assure themselves and readers that their sample reflects average thoracic size and shape patterns.

4. Since KH2 is an undoubted male Neandertal, some readers might wonder why not compare him to modern human males exclusively. For example, 20 modern human males, rather 10 males and females would have doubled the sample size for the most appropriate comparison?

5. Related to these last two issues, Figure 3 shows that two of the modern human males might actually be outliers along the male PC4 distribution. If so, do these two skew multivariate analyses which can be much more sensitive to outliers than univariate analyses? This issue arises again in almost all of the PCA plots presented in the Supplementary Materials where at least one male appears as an obvious outlier. How much does he affect the results? If one or two males are outliers, then the already small sample of $n=10$ becomes even smaller potentially exacerbating further some of the issues raised above. I leave it to the authors for how to address these various issues, but clearly some discussion is warranted.

Finally, I have one last general interpretive issue that I would like to see the authors address, or at least mention, which involves assumptions about the isomorphic nature of thoracic size, lung size, and pulmonary capacity that pervades in the paleoanthropology literature.

Some years back, I was discussing this issue with a seasoned researcher in clinical pulmonary function, and he surprised me by indicating that standard spirometric measures (forced vital capacity, forced expiratory and inspiratory flow, peak expiratory flow, tidal volume flow, total lung capacity, diffusing capacity, max voluntary ventilation, etc.) often showed little to no correlation with overall chest size or lung size. The implication was that breathing work and breathing efficiency was actually governed more by individual differences in chemistry (hematologic and serum biochemical values) and other much smaller scale differences that govern oxygen diffusion across the respiratory membrane and into the pulmonary capillaries, as well as converse pattern of carbon dioxide diffusion. Such factors are unlikely to be reflected in gross thoracic shape and size. Should we be more cautious about the assumption of thoracic size, lung size, and pulmonary capacity?

Reviewer #2:

Remarks to the Author:

Review of Gomez-Olivencia et al.

This ms. would be a welcome addition to the literature, and involves the analysis of data that are really hard to acquire, both in fossil and modern samples. That alone is reason enough to warrant its publication. The authors are also to be commended for situating their study within the published literature - they have been very thorough in this regard.

Their results are a bit surprising, given all the work done through the years on Neanderthal

thoracic elements (and their appendicular proxies such as clavicular length).

A couple of questions occurred to me as I was reading the ms., and I would like to see the authors address these before publication.

First, would it not be desirable to estimate thoracic volume from these data? I understand that the height / resting position of the diaphragm (i.e., the floor of the thoracic cavity) cannot be estimated, but would it not be possible to measure the volume enclosed within these virtual models, even if said volume included some of the abdominal cavity? I wonder if, even though the centroids are not significantly different between modern human males and K2, if there might be a difference in thoracic volume, however estimated between the two taxa?

Second, I understand that the PCA plot in Figure 3 was chosen because it differentiates K2 from the modern human males, but when one has to go to PC3 and PC4 to distinguish K2 from the recent humans (with only ca. 17.7% of the variance explained by these two components), is this really telling us something significant? Was the scatter plot of PC1 and PC2 that uninformative?

Maybe I am talking out of both sides of my mouth, simultaneously doubting that Neanderthals have lung volumes indistinguishable from modern human males while at the same time complaining that the only scatter plot distinguishing the two taxa is one of two principal components that explain very little of the total variance.

Is it not possible that European Neanderthals had more voluminous thoraces than their Levantine cousins? Might this be why K2 is not as different from modern humans as previous analyses have suggested?

MINOR COMMENTS:

Abstract:

"indistinguishable to significantly different from that" would work better as
"indistinguishable from to significantly different than that"

line 61 - "body mass" needs to be two words here.

line 67 - "relying more on the diaphragm contraction" would work better as "relying more on diaphragmatic contraction" (I would change this in line 424, as well)

Introduction:

line 85 - "correlated with" sounds better to me, as does "physiological variable"

line 107 - "humans" should be "human"

The Kebara 2 individual:

line 149 - I do not know what "(60-65 m.a.s.l.)" means

Reviewer #3:

Remarks to the Author:

The Neanderthal thorax was difficult to assess until now due to the fact that it is a complex system composed by a lot of bones. The authors have done a huge work to generate 3D images and to deliver a reconstruction putting together all the bones in an anatomical connection. Thanks to this work, it is possible to look at the complete trunk of the Kebara 2 fossil and to compare it with modern humans.

In order to improve the article, I advise to the authors:

- To summarize the SI in the article explaining how did they do to virtually put the bones in an anatomical connection and how accurate such step is? Dealing with dry bones, it is not easy to put the bones in a "living position". Following this idea, it would be useful to add a picture of the fossil itself as it is preserved.
- To justify the sex estimation of the Kebara 2 fossil (i.e. estimated as a male). Even it is based on the publication by J. Rak (1990), it's good to explain how it was done (I guess on the pelvis, which is the most appropriate bone). Due to the state of preservation of the specimen (as explained in SI), some uncertainties can stay on the pelvis. Results of the analysis of the article are not so convincing in this aspect. Indeed, Kebara 2 is closed to the males from the modern humans group on a PCA (suppl. Fig.3) which takes into account the size effect. In my opinion, the big size of the thorax of a fossil (belonging to an extinct species) is not sufficient to conclude on a sexual determination.
- To avoid sentence as "the pattern shown by the Kebara 2 reconstruction can be extended to other Neandertals". Indeed, it is always risky to enlarge such conclusion based on results obtained on one specimen (i.e. the Kebara 2 fossil) above all in the case of the thorax which is not well preserved in the fossil record leading to an under-estimation of the morphological variability within this group.
- To do minor modifications: 1/ to put in a same color all the bones from Kebara 2 in figures 1 and 2 or to explain the meaning of the 2 colors used, 2/ to add in Table 1 the estimation on stature based on the humerus.

Reviewer 1 – Comments on Article File

Page: 3

Author: redacted Subject: Sticky Note Date: 6/2/2018 12:04:39 PM

While this difference is only slightly larger based on the location it was taken (at the sternal end of Rib 5), The difference at the level of Ribs 7 & appears more substantial as shown in Figure 2. This seems to be contradicted by Figure 2, which shows it to be considerably more A-P enlarged? Also, the legend for this figure actually says: "In lateral view K2 shows relatively larger antero-posterior mid-thorax." Moreover, as noted below, there is clearly one modern human male that is an outlier, and may be slightly biasing this difference (see below).

Author: redacted Subject: Highlight Date: 6/1/2018 2:56:17 PM

Author: redacted Subject: Sticky Note Date: 6/1/2018 1:14:07 PM

What does "height" mean here? Does it mean location as in higher up in the thorax?

Author: redacted Subject: Highlight Date: 6/1/2018 1:13:09 PM

Page: 5

 Author: redacted Subject: Sticky Note Date: 6/1/2018 1:29:14 PM

I would replace this term with "movement" here because "locomotion" in the general sense refers to translocation of an entire organism from one place to another. I realize that in many non-human primates the upper limbs are used to transport the entire organism through an arboreal environment, but in extant humans and closely related species in Homo such as Neandertals, this is not the case. So, as written, the reader will immediately wonder which parts of the thoracic musculature contribute to lower limb movement?

 Author: redacted Subject: Highlight Date: 6/1/2018 1:19:17 PM

 Author: redacted Subject: Sticky Note Date: 6/1/2018 1:41:22 PM

Awkward phrase. Perhaps replace with: "were not, on their own, evidence for a"

 Author: redacted Subject: Highlight Date: 6/1/2018 1:41:41 PM

Page: 6

Author: redacted Subject: Highlight Date: 6/1/2018 2:05:36 PM

Author: redacted Subject: Sticky Note Date: 6/1/2018 2:09:42 PM

While, as written, one can clearly infer the meaning, it sounds a bit like an incomplete sentence.

Author: redacted Subject: Sticky Note Date: 6/1/2018 2:21:33 PM

As written, this sentence and the preceding one appear as disconnected thoughts. Perhaps begin this sentence with: "Sawyer and Maley (2005) also indicated..."

Author: redacted Subject: Highlight Date: 6/1/2018 2:18:43 PM

Author: redacted Subject: Highlight Date: 6/1/2018 2:22:13 PM

Page: 7

Author: redacted Subject: Sticky Note Date: 6/1/2018 2:23:27 PM
Change to: "spinal morphology of Neandertals"

Author: redacted Subject: Sticky Note Date: 6/1/2018 2:34:23 PM
Perhaps should change to: "interpretative" instead.

'Paradigm shifts' in science generally refer to major, overarching, conceptual changes in how questions are asked, or in how data are interpreted. If reinterpretation of Neanderthal posture had resulted in seeing them as poorly-adapted bipeds, for example, then that might rise to a paradigm shift. But clearly, that is not the case.

Author: redacted Subject: Highlight Date: 6/1/2018 2:22:13 PM

Author: redacted Subject: Highlight Date: 6/1/2018 2:25:16 PM

Author: redacted Subject: Sticky Note Date: 6/1/2018 2:35:23 PM
Perhaps should insert "previous" here.

Author: redacted Subject: Highlight Date: 6/1/2018 2:34:59 PM

Author: redacted Subject: Sticky Note Date: 6/1/2018 2:41:55 PM
Since this is the only time it is used in the manuscript, perhaps writing it out as "meters above sea level" is better.

Author: redacted Subject: Highlight Date: 6/1/2018 2:39:44 PM

Author: redacted Subject: Sticky Note Date: 6/1/2018 2:42:32 PM
Change to: "burial", or "inhumation"

Author: redacted Subject: Highlight Date: 6/1/2018 2:42:06 PM

Author: redacted Subject: Sticky Note Date: 6/1/2018 2:43:20 PM
Change to: "The"

Author: redacted Subject: Highlight Date: 6/1/2018 2:43:05 PM

Page: 8

Author: redacted Subject: Sticky Note Date: 6/1/2018 2:44:42 PM
Change to: "at"

Author: redacted Subject: Highlight Date: 6/1/2018 2:44:22 PM

Author: redacted Subject: Sticky Note Date: 6/1/2018 2:46:04 PM
Delete.

Author: redacted Subject: Highlight Date: 6/1/2018 2:45:55 PM

Author: redacted Subject: Sticky Note Date: 6/1/2018 2:46:50 PM
Insert: "an"

Author: redacted Subject: Highlight Date: 6/1/2018 2:46:23 PM

Author: redacted Subject: Sticky Note Date: 6/1/2018 2:51:38 PM
Perhaps "pseudoarthrosis" is better.

Author: redacted Subject: Highlight Date: 6/1/2018 2:51:09 PM

Page: 14

Author: redacted Subject: Sticky Note Date: 6/1/2018 3:06:23 PM
STOPPED HERE.

Author: redacted Subject: Highlight Date: 6/1/2018 3:06:13 PM

Page: 17

Author: redacted Subject: Sticky Note Date: 6/1/2018 5:47:15 PM
Curvatures of what?

Author: redacted Subject: Highlight Date: 6/1/2018 5:46:47 PM

Page: 19

Author: redacted Subject: Sticky Note Date: 6/1/2018 5:49:54 PM
Incidence of what?

Author: redacted Subject: Highlight Date: 6/1/2018 5:49:39 PM

Author: redacted Subject: Sticky Note Date: 6/1/2018 5:50:41 PM
Insert "and" here.

Author: redacted Subject: Highlight Date: 6/1/2018 5:50:22 PM

Page: 21

Author: redacted Subject: Highlight Date: 6/1/2018 3:11:51 PM

Author: redacted Subject: Sticky Note Date: 6/1/2018 3:42:22 PM

Obviously, the entire analysis depends on the degree to which this sample of males and females accurately reflect average modern human thoracic anatomy. Therefore, several questions arise. First, what are the ages of these individuals? Are they all elderly? Or, do they represent a range of adult ages? This is important since advanced age can significantly alter thoracic anatomy due to several hard and soft-tissue changes. Second, what do we know about the overall health of these individuals? Could any of them have had health issues that could have potentially affected their thoracic anatomy? Even in the absence of obvious pathology or extreme age-related changes, on what basis do we know that these 10 males and 10 females reflect average human thoracic size and shape dimensions? Third, since KH2 is a male Neandertal, why not compare him to modern human males exclusively. 20 modern human males, rather 10 males and females would have doubled the sample size for the most appropriate comparison? Related to this last issue, Figure 3 shows that two of the modern human males might actually be outliers along the PC4 distribution. If so, do these two skew multivariate analyses which can be much more sensitive to outliers than univariate analyses? In short, there are a lot of unanswered questions here.

Page: 22

Author: redacted Subject: Sticky Note Date: 6/1/2018 5:36:23 PM

Was this the average of all males? In the early part of the paper, it is clear that comparisons are against males, which is appropriate rather than the sex combined sample. Need to be clear here as well.

Author: redacted Subject: Highlight Date: 6/1/2018 5:24:38 PM

Reviewers' comments:

Reviewer #1 (Remarks to the Author):

Beyond the occasional editing that I've noted directly on the manuscript, there is very little to quibble with here.

[Notes directly posted on the manuscript by Reviewer #1:]

(Abstract) "breadth, but is only slightly larger antero-posteriorly."

While this difference is only slightly larger based on the location it was taken (at the sternal end of Rib 5), The difference at the level of Ribs 7 & appears more substantial as shown in Figure 2. This seems to be contradicted by Figure 2, which shows it to be considerably more A-P enlarged?

Also, the legend for this figure actually says: "In lateral view K2 shows relatively larger antero-posterior mid-thorax." This comment does not longer apply as the abstract has been reduced considerably and that sentence does not appear in the new version of the manuscript.

Moreover, as noted below, there is clearly one modern human male that is an outlier, and may be slightly biasing this difference (see below).

With the enlarged sample, the individual that in the previous analysis could be considered an outlier, is now well within the comparative sample's variation.

(Abstract) "In this case, differences in the size of soft tissues (such as height of the diaphragm)"

What does "height" mean here" Does it mean location as in higher up in the thorax?

It refers to the location within the thorax, but actually, it would be necessary to have a lower position of the diaphragm within the thorax in order to increase the total lung capacity. We have rephrased this sentence and now it reads "In this case, differences in the size and location of soft tissues (such as a lower position of the diaphragm within the thorax) could have resulted in a larger total lung capacity despite a thorax of overall similar size to that of modern humans."

(Introduction) "The thorax is the insertion point of muscles related to locomotion and mobility of the upper limb"

I would replace this term with "movement" here because "locomotion" in the general sense refers to translocation of an entire organism from one place to another. I realize that in many non-human primates the upper limbs are used to transport the entire organism through an arboreal environment, but in extant humans and closely related species in Homo such as Neandertals, this is not the case. So, as written, the reader will immediately wonder which parts of the thoracic musculature contribute to lower limb movement?

Changed.

(Introduction) "...but were not able to assure if they had a more voluminous thorax compared to modern humans (i.e., Shanidar)"

Awkward phrase. Perhaps replace with: "were not, on their own, evidence for a"

Changed.

(Introduction) "Within the context of the reconstruction of a complete Neandertal 126 skeleton, the thoracic and lumbar spine and costal skeleton of K2 were used."

While, as written, one can clearly infer the meaning, it sounds a bit like an incomplete sentence. We have completed this sentence. Now it reads: “Within the context of the reconstruction of a complete Neandertal skeleton, skeletal elements from La Ferrassie 1 were used combined with those from other Neandertal individuals, e.g. the thoracic and lumbar spine and costal skeleton of K2 were used²⁷.”

(Introduction) “Sawyer and Maley (2005) indicated the presence”

As written, this sentence and the preceding one appear as disconnected thoughts. Perhaps begin this sentence with: "Sawyer and Maley (2005) also indicated..."

Done. Additionally we have changed the reference system and now it reads “Sawyer and Maley²⁷ also indicated...”

(Introduction) “the Neandertal spine morphology which”

Change to: "spinal morphology of Neandertals"

Done.

(Introduction) “a paradigm shift”

Perhaps should change to: "interpretative" instead. 'Paradigm shifts' in science generally refer to major, overarching, conceptual changes in how questions are asked, or in how data are interpreted. If reinterpretation of Neandertal posture had resulted in seeing them as poorly-adapted bipeds, for example, then that might rise to a paradigm shift. But clearly, that is not the case.

Done.

(Introduction) “the reconstruction”

Perhaps should insert "previous" here.

Done.

(The Kebara 2 individual) “The cave site of Kebara (60-65 m.a.s.l.)”

Since this is the only time it is used in the manuscript, perhaps writing it out as "meters above sea level" is better.

Done.

(The Kebara 2 individual) “The sepulture of K2”

Change to: "burial", or "inhumation"

Done.

(The Kebara 2 individual) “This”

Change to: "The"

Done.

(The Kebara 2 individual) “His body mass was estimated to 75.6 kg”

Change to: "at"

Done.

(The Kebara 2 individual) “pseudoarthroses in the ribs 5-7”

Delete

Done.

(The Kebara 2 individual) “by anatomical variant”

Insert: "an"

Done.

(The Kebara 2 individual) “and **nearthrosis** observed”

Perhaps "pseudoarthrosis" is better.

“Nearthrosis” is an accepted medical term for a new joint. A “pseudoarthrosis” would be a kind of nearthrosis as this term refers to a new, false joint arising at the site of an ununited fracture. Thus, we have preferred to retain “nearthrosis”

(Discussion) “responsible for (or, at least, related to) the lower degree of curvatures,”

Curvatures of what?

The curvatures of the spine. Added to the text.

(Discussion) “e.g. lower pelvic incidence, more”

Incidence of what?

“Pelvic incidence” is an anatomical parameter proposed by Lagaye and colleagues and which has been widely used in different papers.

(Discussion) “lumbar lordosis, lateral orientation”

Insert "and" here.

Done.

(Traditional and geometric morphometric analysis) “In order to be conservative and due to the lack of other Neandertal ribcages to use as references, we used the coordinates of the average *Homo sapiens* thorax as a reference specimen.”

Was this the average of all males? In the early part of the paper, it is clear that comparisons are against males, which is appropriate rather than the sex combined sample. Need to be clear here as well.

Yes. Now it reads “we used the coordinates of the mean *Homo sapiens* male thorax as a reference specimen”.

The one aspect of the manuscript that I think can be improved has to do with the presentation and details provided of the extant human sample from which, and against which the Kebara thoracic remains are compared. Obviously, the analysis depends heavily on the degree to which this sample of males and females accurately reflects average modern human thoracic anatomy. Therefore, several questions arise that are not addressed in the materials as presented:

1. What are the ages of these individuals? Are they all elderly? Or, do they represent a larger range of adult ages? This is potentially important since advanced age can significantly alter prime-age thoracic anatomy due to several hard and soft-tissue changes.

We now include the age/age-at-death of the modern comparative sample in Table 1. Despite the differences in age (or age-at-death) between K2 and this modern human sample (Table 1), we consider this comparative sample to be representative of adult modern males, as a comparison of this sample to that of another adult modern male sample (n=18; age = 51.9 +/- 1.2 years; García-Martínez et al., 2016) using a landmark protocol of 402 landmarks (ribs 1-10) did not yield statistical differences between the samples (permutation test; n=100; p=0.14).

2. What do we know about the overall health of these individuals? Could any of them have had health issues that could have potentially affected their thoracic anatomy?

All individuals were scanned in supine position and we avoided CT-scanning individuals with significant morphological deformations and/or causes of death that might have altered ribcage morphology. This is now clearly stated in the material and methods section.

3. Even in the absence of obvious pathology or extreme age-related changes (if that is the case), on what basis do we know that these 10 males and 10 females reflect average human thoracic size and shape dimensions? Some large research universities with medical schools in which pulmonary health and physiology is studied have developed, or have access to CT-scan studies that have provided a sample of “normals” against which their clinical patients can be compared. The authors of the current manuscript have done a lot of work, and I am not asking them to re-do their analysis. I am just wondering if there is a way for them to compare their particular sample to a set of such normals, even in a cursory fashion with a few key variables from the published literature, to assure themselves and readers that their sample reflects average thoracic size and shape patterns.

This is answered in question 1.

4. Since KH2 is an undoubted male Neandertal, some readers might wonder why not compare him to modern human males exclusively. For example, 20 modern human males, rather 10 males and females would have doubled the sample size for the most appropriate comparison?

We have increased as much as we could the modern human male sample, and now it is composed of 16 males.

5. Related to these last two issues, Figure 3 shows that two of the modern human males might actually be outliers along the male PC4 distribution. If so, do these two skew multivariate analyses which can be much more sensitive to outliers than univariate analyses? This issue arises again in almost all of the PCA plots presented in the Supplementary Materials where at least one male appears as an obvious outlier. How much does he affect the results? If one or two males are outliers, then the already small sample of $n=10$ becomes even smaller potentially exacerbating further some of the issues raised above. I leave it to the authors for how to address these various issues, but clearly some discussion is warranted.

This comment by the reviewer does not longer apply. Now, with an increased sample of 16 males and the multivariate analysis focused only on male specimens, no modern human from the sample seems to be an outlier.

Finally, I have one last general interpretive issue that I would like to see the authors address, or at least mention, which involves assumptions about the isomorphic nature of thoracic size, lung size, and pulmonary capacity that pervades in the paleoanthropology literature.

Some years back, I was discussing this issue with a seasoned researcher in clinical pulmonary function, and he surprised me by indicating that standard spirometric measures (forced vital capacity, forced expiratory and inspiratory flow, peak expiratory flow, tidal volume flow, total lung capacity, diffusing capacity, max voluntary ventilation, etc.) often showed little to no correlation with overall chest size or lung size. The implication was that breathing work and breathing efficiency was actually governed more by individual differences in chemistry (hematologic and serum biochemical values) and other much smaller scale differences that govern oxygen diffusion across the respiratory membrane and into the pulmonary capillaries, as well as converse pattern of carbon dioxide diffusion. Such factors are unlikely to be reflected in gross thoracic shape and size. Should we be more cautious about the assumption of thoracic size, lung size, and pulmonary capacity?

The reviewer raises an important issue here. We have included a cautionary note just briefly addressing this issue: an additional phrase at the end of the “Was there a different breathing mechanism in Neandertals?” section of the Discussion.

Reviewer #2 (Remarks to the Author):

Review of Gomez-Olivencia et al.

This ms. would be a welcome addition to the literature, and involves the analysis of data that are really hard to acquire, both in fossil and modern samples. That alone is reason enough to warrant its publication. The authors are also to be commended for situating their study within the published literature - they have been very thorough in this regard.

Their results are a bit surprising, given all the work done through the years on Neanderthal thoracic elements (and their appendicular proxies such as clavicular length).

A couple of questions occurred to me as I was reading the ms., and I would like to see the authors address these before publication.

First, would it not be desirable to estimate thoracic volume from these data? I understand that the height / resting position of the diaphragm (i.e., the floor of the thoracic cavity) cannot be estimated, but would it not be possible to measure the volume enclosed within these virtual models, even if said volume included some of the abdominal cavity? I wonder if, even though the centroids are not significantly different between modern human males and K2, if there might be a difference in thoracic volume, however estimated between the two taxa?

We consider that the volume would be another proxy for size, and we have already provided with different proxies for size, such as the centroid size (CS) and different traditional measurements of the reconstructed thorax compared to our modern male sample. The geometric morphometric approach is consistent with the traditional measurements because, despite the fact that K2 shows a significantly wider thorax, the height is smaller and hence the similar CS of K2 compared to our modern human comparative sample.

Second, I understand that the PCA plot in Figure 3 was chosen because it differentiates K2 from the modern human males, but when one has to go to PC3 and PC4 to distinguish K2 from the recent humans (with only ca. 17.7% of the variance explained by these two components), is this really telling us something significant? Was the scatter plot of PC1 and PC2 that uninformative?

This comment no longer applies as the new Figure 3 shows the PC2 vs PC3, in which we compare Kebara 2 to only modern males (as requested by Reviewer #1). The differences are significant based on the permutation test which shows significant differences between K2 and the modern human male comparative sample. The new scatter plot of PC1 and PC2 is included in the supplementary information and it is not very informative. We interpret these results as Neandertals having a general thorax shape similar to that of other members of genus *Homo*. For example, Neandertals would group with modern humans if African apes were included in the PCA. Nonetheless, K2 is still significantly different from modern human males.

Maybe I am talking out of both sides of my mouth, simultaneously doubting that Neanderthals have lung volumes indistinguishable from modern human males while at the same time complaining that the only scatter plot distinguishing the two taxa is one of two principal components that explain very little of the total variance.

Is it not possible that European Neanderthals had more voluminous thoraces than their Levantine cousins? Might this be why K2 is not as different from modern humans as previous analyses have suggested? This possibility was proposed by Franciscus and Churchill in 2002 comparing the first two ribs of Regourdou and La Chapelle-aux-Saints 1 (LC1) to that of Kebara 2. Afterwards Gómez-Olivencia et al. (2009), showed that:

-1) the first rib of Amud 1 could have been larger than that of Regourdou 1

-2) the anatomical orientation of the LC1 second rib was wrong and thus the argument based on this rib was no longer useful.

At this point, no evidence supports this idea of differences between Levantine and European Neandertals in their thorax morphology. Unfortunately, the current Neandertal thorax fossil record is too sparse to properly address this hypothesis, analysis of which should also take into consideration chronological (and thus climatic) issues.

MINOR COMMENTS:

Abstract:

“indistinguishable to significantly different from that” would work better as “indistinguishable from to significantly different than that” Changed.

line 61 - “body mass” needs to be two words here.

Changed.

line 67 - “relying more on the diaphragm contraction” would work better as “relying more on diaphragmatic contraction” (I would change this in line 424, as well)

Changed.

Introduction:

line 85 - “correlated with” sounds better to me, as does “physiological variable”

Changed.

line 107 - “humans” should be “human”

Changed.

The Kebara 2 individual:

line 149 - I do not know what “(60-65 m.a.s.l.)” means

Now it reads “meters above sea level” following the suggestion made by Reviewer #1.

Reviewer #3 (Remarks to the Author):

The Neanderthal thorax was difficult to assess until now due to the fact that it is a complex system composed by a lot of bones. The authors have done a huge work to generate 3D images and to deliver a reconstruction putting together all the bones in an anatomical connection. Thanks to this work, it is possible to look at the complete trunk of the Kebara 2 fossil and to compare it with modern humans.

In order to improve the article, I advise to the authors:

- To summarize the SI in the article explaining how did they do to virtually put the bones in an anatomical connection and how accurate such step is? Dealing with dry bones, it is not easy to put the bones in a “living position”.

The virtual reconstruction is a methodology that has seen widespread use in the field and thus we consider that the basic steps to have been well-explained elsewhere (e.g., the book “Virtual Anthropology: A Guide to a New Interdisciplinary Field” or Suwa et al., 2009-Science). The information provided in the SI cannot properly be summarized in the main text. Moreover, there is not enough space, in order to comply to the 5,000 word limit of the Nature communications format requirements and, thus, we prefer to keep it in the SI. In fact, we have clearly stated the possible inaccuracies of the reconstruction and how we have tried to overcome them.

Following this idea, it would be useful to add a picture of the fossil itself as it is preserved.

We have added several pictures in the supplementary information:

1-Photograph of the cast of the Kebara 2 skeleton as it was found which shows the preserved anatomical elements

2-Cranial view of all the thoracic vertebrae

3-Cranial view of all the costal remains

4-Ventral view of the sternal remains

- To justify the sex estimation of the Kebara 2 fossil (i.e. estimated as a male). Even it is based on the publication by J. Rak (1990), it's good to explain how it was done (I guess on the pelvis, which is the most appropriate bone). Due to the state of preservation of the specimen (as explained in SI), some uncertainties can stay on the pelvis. Results of the analysis of the article are not so convincing in this aspect. Indeed, Kebara 2 is closed to the males from the modern humans group on a PCA (suppl. Fig.3) which takes into account the size effect. In my opinion, the big size of the thorax of a fossil (belonging to an extinct species) is not sufficient to conclude on a sexual determination.

The sex assessment of Kebara 2 was based on the pelvic morphology and it is now more clearly stated on the text. Kebara 2 is regarded as a male individual in the paleoanthropological community (see comments by Reviewer #1 or Ruff et al., 1997-Nature). Based on the suggestions made by the reviewer 1 we now only compare Kebara 2 to male individuals. Based on the results from the previous version of the manuscript and the new analysis with the enlarged male sample the taxonomic differences between Kebara 2 (as a representative of Neandertals) and modern humans are larger than the within modern human morphological variation due to sexual dimorphism. Moreover, regardless of the sex of Kebara 2, comparing it only to males means that we compare to modern humans who are more similar to him in stature (see Table 1) which makes the analysis more conservative and sensitive to taxonomic, rather than size-based, differences.

- To avoid sentence as “the pattern shown by the Kebara 2 reconstruction can be extended to other Neandertals”. Indeed, it is always risky to enlarge such conclusion based on results obtained on one specimen (i.e. the Kebara 2 fossil) above all in the case of the thorax which is not well preserved in the fossil record leading to an under-estimation of the morphological variability within this group. We have slightly changed the wording of this sentence and now it reads: “Further, we hypothesize that the...”. We consider this study important not only because this is the first time that we can accurately reconstruct a Neandertal thorax but also because the reconstruction of the Kebara 2 Neandertal is informative to assess the paleobiology of Neandertals as a group.

- To do minor modifications: 1/ to put in a same color all the bones from Kebara 2 in figures 1 and 2 or to explain the meaning of the 2 colors used,2/ to add in Table 1 the estimation on stature based on the humerus. Done

Additional changes:

1-The abstract has been reduced to 150 words to comply with the Nature communications format requirements.

2-The subsection headings have been removed from the Introduction section to comply with the Nature communications format requirements.

3-The subsection headings have been removed from the Discussion section to comply with the Nature communications format requirements.

REVIEWERS' COMMENTS:

Reviewer #1 (Remarks to the Author):

I am pleased to see that the authors, in this version, have improved their comparative sample in quantity and quality as I requested, and that they have now also provided the necessary detail to help readers assess the veracity of their comparative framework and thus their results. While the age-at-death difference between their improved sample and KH2 is considerable (and still gives me pause), they have in this revised manuscript also provided comparisons with a second younger sample, as I requested, which definitely strengthens the manuscript. Among the many improvements that their enhanced sampling protocol has yielded is the disappearance of one or two outliers that previously cast some doubt on the reliability of the results. I find all of these changes to be very positive, and I congratulate the authors for dealing with these issues effectively despite the extra work that it required.

The authors have also edited the manuscript grammatically as requested. I have only a few minor suggestion regarding editing with the current manuscript:

1. In the Abstract, the first sentence is awkwardly phrased. Currently it reads: "The size and shape of the Neandertal thorax has been the object of debate since the first discovery of Neandertal ribs more than 150 years ago and have proposed different morphologies, ranging from a Neandertal morphology indistinguishable from to significantly different than that of modern humans." Change to: "The size and shape of the Neandertal thorax has been debated since the first discovery of Neandertal ribs more than 150 years ago with workers proposing different interpretations ranging from a Neandertal thoracic morphology that is indistinguishable from modern humans, to one that was significantly different from them."

2. Also, in the Abstract, the last sentence can be improved both grammatically and factually: Currently, it reads: "Nevertheless, kinematic analyses show that rib cages which are wider in its lower segment produce greater overall size increments (respiratory capacity) during inspiration. We hypothesize that Neandertals may have had a somewhat different breathing mechanism. Change to: "Nevertheless, kinematic analyses show that rib cages which are wider in their lower segment produce greater overall size increments (respiratory capacity) during inspiration. We hypothesize that Neandertals may have had a subtle, but somewhat different breathing mechanism."

3. Page 13, line 295: The first clause in this sentence is incomplete and incomprehensible: it currently reads: "Neandertals are significantly diff average Homo sapiens sis," This must be a cutting and pasting error, and needs to be fixed.

4. Page 14, the sentence beginning on line 313 currently reads: "In summary, the present reconstruction demonstrates that significant differences exist in the thorax shape within genus Homo." I would change this to read: "In summary, the present reconstruction demonstrates that subtle, but significant differences exist in the thorax shape within genus Homo." I also suggested the insertion of the term "subtle" in the abstract (point 1 above)

for the same reason: namely, while this work shows statistically significant anatomical differences with the comparative sample they use, this is different from the broader interpretation that the breathing physiology overall is significantly different from other members of the genus Homo. By inserting the term "subtle" they are more properly contextualizing their results.

With these minor suggested comments noted, I find this revised manuscript now suitable for publication.

Reviewer #2 (Remarks to the Author):

It was very useful to read the comments of the other two referees. What is apparent is that the authors have taken the suggestions of all three of us into account, and as a result this is a much improved manuscript.

I believe it to be now ready for publication.

Reviewer #3 (Remarks to the Author):

Thank-you for taking my suggestions into account. This manuscript in my opinion is now ready to be published.

REVIEWERS' COMMENTS:

Reviewer #1 (Remarks to the Author):

I am pleased to see that the authors, in this version, have improved their comparative sample in quantity and quality as I requested, and that they have now also provided the necessary detail to help readers assess the veracity of their comparative framework and thus their results. While the age-at-death difference between their improved sample and KH2 is considerable (and still gives me pause), they have in this revised manuscript also provided comparisons with a second younger sample, as I requested, which definitely strengthens the manuscript. Among the many improvements that their enhanced sampling protocol has yielded is the disappearance of one or two outliers that previously cast some doubt on the reliability of the results. I find all of these changes to be very positive, and I congratulate the authors for dealing with these issues effectively despite the extra work that it required.

The authors have also edited the manuscript grammatically as requested. I have only a few minor suggestions regarding editing with the current manuscript:

1. In the Abstract, the first sentence is awkwardly phrased. Currently it reads: “The size and shape of the Neandertal thorax has been the object of debate since the first discovery of Neandertal ribs more than 150 years ago and have proposed different morphologies, ranging from a Neandertal morphology indistinguishable from to significantly different than that of modern humans.” Change to: “The size and shape of the Neandertal thorax has been debated since the first discovery of Neandertal ribs more than 150 years ago with workers proposing different interpretations ranging from a Neandertal thoracic morphology that is indistinguishable from modern humans, to one that was significantly different from them.” Done

2. Also, in the Abstract, the last sentence can be improved both grammatically and factually: Currently, it reads: “Nevertheless, kinematic analyses show that rib cages which are wider in its lower segment produce greater overall size increments (respiratory capacity) during inspiration. We hypothesize that Neandertals may have had a somewhat different breathing mechanism. Change to: “Nevertheless, kinematic analyses show that rib cages which are wider in their lower segment produce greater overall size increments (respiratory capacity) during inspiration. We hypothesize that Neandertals may have had a subtle, but somewhat different breathing mechanism.” Done

3. Page 13, line 295: The first clause in this sentence is incomplete and incomprehensible: it currently reads: “Neandertals are significantly diff average Homo sapiens sis,” This must be a cutting and pasting error, and needs to be fixed. A part of the text from the first version was lost in a cut & paste error. This error has been fixed.

4. Page 14, the sentence beginning on line 313 currently reads: “In summary, the present reconstruction demonstrates that significant differences exist in the thorax shape within genus Homo.” I would change this to read: “In summary, the present reconstruction demonstrates that subtle, but significant differences exist in the thorax shape within genus Homo.” I also suggested the insertion of the term “subtle” in the abstract (point 1 above) for the same reason: namely, while this work shows statistically significant anatomical differences with the comparative sample they use, this is different from the broader interpretation that the breathing physiology overall is significantly different from other members of the genus Homo. By inserting the term “subtle” they are more properly contextualizing their results. Done

With these minor suggested comments noted, I find this revised manuscript now suitable for publication.

Reviewer #2 (Remarks to the Author):

It was very useful to read the comments of the other two referees. What is apparent is that the authors have taken the suggestions of all three of us into account, and as a result this is a much improved manuscript.

I believe it to be now ready for publication.

Reviewer #3 (Remarks to the Author):

Thank-you for taking my suggestions into account. This manuscript in my opinion is now ready to be published.